# An Experimental Study on the Solidification Treatment of Debris Flow Siltation

**DOI:** 10.3390/ma15196860

**Published:** 2022-10-02

**Authors:** Fengyu Gu, Linrong Xu, Na Su

**Affiliations:** 1School of Civil Engineering, Central South University, Changsha 410075, China; 2National Engineering Laboratory for High Speed Railway Construction, Central South University, Changsha 410075, China

**Keywords:** debris flow, siltation, reinforcement treatment, strength, microscopic mechanism

## Abstract

In recent years, the resulting siltation from railway debris flow disasters has seriously affected the normal use of railway traffic lines and brought great challenges to rescue work. In view of this, we used an orthogonal test scheme to prepare different types of debris flow accumulation and carried out penetration resistance tests in order to explore the effects of different types of curing agents, the amount of curing agent added, the moisture content of debris flow siltation, and the grain gradation of debris flow sediment on the solidification strength of debris flow siltation. We also utilized scanning electron microscopy (SEM) to observe the microstructure and potential curing mechanism of the samples treated with different curing agents in attempt to discern the reasons for their different levels of strength. Our results show that the each of four curing agents tested can effectively improve the solidification strength of the siltation. Furthermore, we found that the type of curing agent had the largest impact on the curing strength of the siltation, followed by the moisture content of the siltation itself, the amount of curing agent added, and particle size. To achieve the best results, we recommend using 14% sulfoaluminate cement as the curing agent.

## 1. Introduction

Debris flow is a common adverse geological phenomenon in mountainous areas [1,2,3] and frequently occurs suddenly and violently, often resulting in widespread disasters [4], especially for railway lines. Railway lines most commonly experience channel-type debris flow gullies [5]. Once a debris flow breaks out, extensive siltation forms in these gullies, and this seriously affects the normal passage of railway traffic and can even lead to railway traffic accidents [5]. Due to the low strength and wide coverage of the siltation itself [6,7,8], it is also difficult to carry out rescue and relief work when such accidents occur. In recent years, railway traffic accidents caused by debris flow have occurred quite frequently. Therefore, the problem of restoring the smoothness of railways as soon as possible after railway traffic is buried in debris flow is an urgent one [9,10]. To this end, solidification of railway debris flow siltation can facilitate the intervention of rescue personnel and equipment and better protect the lives of the trapped people. Hence this study examines the best ways to achieve this solidification.

At present, the research on debris flow siltation primarily focuses on the deposition range and deposition characteristics of debris [11,12,13,14,15,16,17,18,19,20,21]. Tian et al. analyzed the submerged range of the debris flow siltation in the Qinglin Basin through numerical simulation [22], and Vosoughi et al. studied the depth of debris flow siltation through field observation as well as numerical modeling [23]. Similarly, Zhou et al. analyzed the effects of water content and slope on the sedimentary characteristics of debris flow siltation through experimental tests [24]. However, there are few studies on the solidification of debris flow siltation. Thus, this paper aims to study the solidification treatment of debris flow siltation by considering the influence of different factors on the solidification strength of railway debris flow siltation, making use of orthogonal testing in particular. The study also analyzes the microscopic solidification mechanism of debris flow siltation under the action of different curing agents in order to discern the reasons for their different curing effects. Our research results can provide theoretical and technical reference for the solidification treatment of debris flow siltation for railway traffic rescue engineering in the future.

## 2. Materials and Methods

### 2.1. Experimental Materials

The debris material for our experiments was taken from Changsha, Hunan Province, China. Before our laboratory testing, the soil samples in the material need to be fully aired and dried, and they need to be crushed to prevent clumping; then the material is sieved, and particles larger than 1 cm were eliminated. The particle sizes of the soil sample were then divided into three particle groups: silt clay (<0.075 mm), sandy soil (0.075 to 2 mm), and fine gravel soil (2 to 10 mm), according to our field investigation and the related literature [25,26]. Different debris flow siltations were prepared from soil samples according to our orthogonal test design. The samples (without curing agents) were qualitatively tested by X-ray diffractometer (Research Institute of Mining and Metallurgy, Changsha, China), and the test results showed that the crystalline composition of the samples was mostly quartz, feldspar, chlorite, calcite, and mica (Figure 1). We considered four kinds of curing agents in this study, sulfoaluminate cement (Jiuqi Building Materials, R.SAC42.5, Yantai, China), Portland cement (Shuoshun Building Materials, P.O42.5, Jinan, China), Sulfoaluminate cement and lime (3%) (Beichen Founder, Jinan, China), Portland cement and lime (0.3%); the experimental water is tap water.

### 2.2. Experimental Methods

#### 2.2.1. Experimental Design

For this study we adopted a 4-factor, 4-level orthogonal design, and the test table is L_16_(4^4^) (Table 1), where columns A through C are the type of curing agent, the amount of curing agent added, and the moisture content, respectively. Column D is the particle gradation of the debris flow siltation. Refer to the relevant literature for more information [25,26]. Particle gradation is divided into 4 levels: gradation 1: 78% fine gravel (2 to 10 mm), 18% sandy soil (0.075 to 2 mm)%, and 4% silt clay (<0.075 mm); gradation 2: 58% fine gravel, 36% sandy soil, and 6% silt clay; gradation 3: 38% fine gravel, 54% sandy soil, and 8% silt clay; and gradation 4: 18% fine gravel, 72% sandy soil, and 10% silt clay. The corresponding relationship between levels and factors is shown in Table 1 using a Concrete Penetration Resistance Tester (Changsha, China) to determine specimen strength and using this penetration resistance value to characterize the corresponding solidification strength value of the debris flow siltation. Considering that there is typically a 72-h “golden” rescue window in order to allow for follow-up rescue work after initial debris flow siltation curing, we used the penetration resistance value at 1d as the final test index.

#### 2.2.2. Experimental Procedure

According to the arrangement of orthogonal experiment, we prepared several groups of debris flow siltation samples, and different curing agents were added according to its requirements. For each sample, we fully stirred the soil-curing-agent mixture, recorded the completion time of sample preparation, and immediately used a penetration resistance meter to measure the resistance value of each sample after the samples had cured for 1 day, the type of penetration resistance meter used is HG-1000, the diameter of the probe is 8 mm, and the effective length is 25 mm. The operation steps are as follows: the debris flow siltation samples were placed in a volumetric bucket (3 L), and the penetration resistance value of each sample was measured with a permeability resistance meter immediately after the samples were cured for 1 day. The tip of the probe was kept level with the sample plane, and the force was applied slowly and evenly until the maximum depth of the probe was reached, and the maximum resistance value during this process was recorded. Each test was performed in triplicate. The average resistance value of the three samples was then used as the final value. After this, we sampled each test piece, completely immersed the samples in ethanol for preservation and to prevent them from continuing to hydrate, then sent them to the testing laboratory for drying. Here, the samples were ground and sprayed with gold in order to observe their morphology better and to eliminate the electric charge effects when observing nonconductive samples by scanning electron microscope (SEM) (Research Institute of Mining and Metallurgy, Changsha, China), which was carried out to detect microscopic structural features.

## 3. Results and Analysis

### 3.1. Orthogonal Test Results

Strength tests were carried out according to the orthogonal scheme, and we also performed visual, range, and variance analyses on the data separately in order to explore the influence of different factors on the strength of the sample and then find the optimal siltation solidification treatment scheme.

#### 3.1.1. Intuition and Range Analysis of Orthogonal Test

Our intuitive analysis results are shown in Table 1. By calculating the sum of the test indices (*k*_1*j*_, *k*_2*j*_, *k*_3*j*_, *k*_4*j*_) and the average test indices ($\bar{k_{1j}}$, $\bar{k_{2j}}$, $\bar{k_{3j}}$, $\bar{k_{4j}}$) of each factor corresponding to the same level, the range R of each column can be calculated so that the order of importance of the factors can be determined; a large range indicates that the factor is important. From Table 1, we can see that the order of importance of the four factors is A > C > B > D. That is, among the four influencing factors, the type of curing agent has the greatest influence, followed by moisture content, the amount of curing agent added, and the particle gradation.

In order to facilitate the analysis of the influence of each factor on curing strength, the data for each factor are graphed in Figure 2, where the ordinate is the average test index [27], and the abscissa is the different penetration resistance values that correspond to each influencing factor. Here we see that the type of curing agent has the greatest influence on the curing strength, and Figure 2a further shows the curing effect of sulfoaluminate cement results in the highest penetration resistance value. Therefore, sulfoaluminate cement should be the first choice as the curing agent for debris flow siltation.

Figure 2b shows that within a certain range, the strength of the samples increases with mixing. Specifically, strength increased the fastest when the dosage increased from 12% to 14%, after which the rate of increase slowed. Thus, 14% appears to be the optimal dosage of curing agent. For water content, Figure 2c shows that (within a certain range) the solidification strength of the siltation gradually decreases with increasing water content. For the range we considered, solidification strength reached its lowest point when the water content was 31%, though this was also the highest water content we tested. Thus, in some cases drainage measures may be necessary to reduce moisture content in order to enhance the curing effects of curing agents in order to increase their strength.

Figure 2d shows that (within a certain range) when the particle gradation of the siltation changes from coarse to fine, its strength decreases. In particular, the sieving test and the liquid-plastic limit test of the samples showed that the gradations (1 through 4) were breccia, breccia, medium sand, and fine sand, respectively. Figure 2d also shows that the solidification of fine sand was the worst and that curing strength decay was the greatest when medium sandy soil changed to fine sandy soil.

#### 3.1.2. ANOVA for the Orthogonal Experiments

The significance of the influence of each factor on the solidification strength of debris flow siltation can be judged by the variance analysis of the results of our orthogonal test. Table 2 shows the results of our variance analysis of the corresponding orthogonal test. Here we can see that factor A has the most significant effect on the strength of solidified debris flow siltation and that the significance level is particularly notable. That is, the type of curing agent has a very large influence on the solidified strength of the debris flow sediment, so it is the first factor to consider in the selection of the overall solidification scheme. After this, factor C, which is moisture content, has a significant influence on curing strength. Compared to these first two factors, factors B and D (dosage of curing agent and particle grain size, respectively) have less influence, but the effect of the amount of curing agent added on the curing strength is greater than the effect of the gradation of the siltation itself. Hence in terms of both priority and operability, the amount of incorporation should be given appropriate consideration. These results agree with our intuitive and range analysis as well.

### 3.2. Microscopic Mechanism Analysis

The above experimental results show that the type of curing agent has the greatest influence on the curing strength of the siltation. Therefore, we prepared the siltation samples under the action of four different curing agents and used the untreated siltation samples as the control group for SEM analysis. As mentioned earlier, SEM analysis was carried out in order to explore the microscopic mechanisms governing the solidification of debris flow siltation under the action of different curing agents and to reveal the reasons for the differences in curing strength between the different curing agents.

Figure 3 is the microscopic image of our silt sample without curing agent treatment, Figure 3a is the joint between the sand-gravel particles and soil in the sample, and Figure 3b is the fine sand-gravel particles covered by soil that are scattered throughout the sample. We can see that there are obvious depressions between the sand and gravel particles and the soil when the sample is not treated with the curing agent, and the porosity between the particles is also large, so an effective connection cannot be established between diverse soil particles. Thus, the soil integrity is poor and its strength is low. Figure 3c is the front view of a depression, and Figure 3d is a partial enlarged view. From these figures we can see that the gravel surface is evenly covered by mud, that the surface is smooth, and that degree of connection between sand and gravel particles and therefore overall strength depend on the degree of natural consolidation of the mud. Hence, the overall structure is poor, and the strength is low as the mud is not consolidated to any appreciable degree.

Figure 4 is a microscopic image of the sample treated with sulfoaluminate cement, and Figure 4a is a morphological diagram of the combination of particles and hydration products in the sample. After treatment, hydration products can be more widely generated, and their overall distribution is uniform. The hydration products are mostly “needle” or “prismatic” fibrous crystals that tightly wrap the soil particles on the surface of the gravel, weaving them into strings and intertwining them into a crystal network structure that causes the soil particles to establish a very effective connection with each other. As a result, soil strength increases rapidly.

Figure 4b shows the combination of sand, gravel particles, and soil. We can see that a large number of evenly distributed hydration products are attached to the surface of the sand and gravel, that the frictional resistance of the surface is greatly increased, and that the cohesion between particles is evidently improved. Soil particles and fine particles form a unified whole with greater strength under the action of hydration products, and the cementation between the sand and gravel particles seems to be more sufficient. Furthermore, the porosity is greatly reduced where the particles are connected to each other, making for a stabler connection, and the improvement in strength is therefore obvious. Figure 4c is a partially enlarged view where we can see that the hydration products appear in the form of fibers or fiber clusters that make the connections between the interior of the sample pack together more closely, thus improving the compressive strength of the sample. In addition, the cement aggregate itself can act as a filler, filling the depressions and pores in the sample and making the overall structure stronger as a result.

Figure 5 is the microscopic image of the sample treated with sulfoaluminate cement lime; Figure 5a,b are the combination of sand, gravel particles, and soil; Figure 5c is the hydration product form at the crack; and Figure 5d is a partial enlargement of the Figure 5c. The difference between Figure 4 and Figure 5 is that the hydration products of the samples treated with sulfoaluminate cement and lime are reduced compared to those in Figure 4, and the hydration products are also fluffier, with shorter lengths. Moreover, some cracks appear locally in Figure 5, and these cracks reduce strength (Figure 5c,d). Figure 5b shows a state of partial suspension at the connection between the gravel particles and the soil. The degree of connection is not as extensive as that of the sample depicted in Figure 4, and the local strength is therefore lower. Additionally, we can see that there are hydration products evenly distributed on the surface of the gravel particles. In the overall structure composed of soil particles and fine particles, the growth of hydration products appears to be good, and their shape is that of a “bird’s nest” shape. Overall, the strength of the samples treated with sulfoaluminate cement and lime decreased compared to that of the sulfoaluminate cement only samples.

Figure 6 is the microscopic image of the sample treated with Portland cement, and Figure 6a is the combination of sand, gravel particles, and soil. Here we can see that these samples also have hydration products after 1 day of curing.

However, compared with sulphoaluminate cement and sulphoaluminate cement with lime, Portland cement samples have fewer and shorter hydration products. Furthermore, the joint surfaces at the meeting of the sand, gravel particles, soil, and hydration products are smaller, and the gap between particles is also smaller. However, the connection is weaker, the porosity is greater, the integrity is worse, and the strength of the sample s is therefore lower overall.

Figure 7 is the microscopic image of the sample treated with Portland cement and lime, and Figure 6a is the joint formed by the sand, gravel particles, and soil. Figure 7b,c are partially enlarged views. From Figure 7d, we see that the samples treated with Portland cement and lime also have hydration products after 1 d of curing, but unlike Portland cement, these hydration products are finer. What’s more, the specific form of these hydration products can be clearly observed only under higher magnification. In addition, some small cracks formed inside the sample, which reduce the connection strength between particles, and the porosity is evidently larger, causing the overall strength of this sample to be lower than that of the sample with Portland cement alone.

The SEM analysis shows that although each of the four different curing agents could improve the strength of the siltation, the sample that used only sulphoaluminate cement performed the best, consistent with the results of our orthogonal testing.

Accordingly, we recommended using 14% sulfoaluminate cement as the curing agent for the practical siltation of debris flow. If conditions permit, rapid drainage measures can also be taken to give full play to the curing effects of this curing agent under reduced moisture content.

## 4. Conclusions

We found that each of the four curing agents we tested can significantly improve the strength of the siltation and that the type of curing agent has the greatest influence on the curing strength of the siltation after 1 day, followed by moisture content, the amount of mixing, and the strength of the gradation. Based on the specific values of the above results, we recommend 14% sulfoaluminate cement as the curing agent. Furthermore, if conditions permit, measures to speed up the drainage should be implemented to take full advantage of the better performance of this curing agent in the presence of a lower moisture environment.

Additionally, when any of the four curing agents were used to treat the siltation, hydration products were formed that were evenly attached to the surface of the particles so that the number of connections between particles grew. This made originally smooth particle surfaces rougher and more multi-faceted and greatly increased the friction between particles, resulting in greater solidification strength. Finally, we found that sulfoaluminate cement had the best curing effect, that its hydration products had the longest length and the widest distribution, that its connection between particles was also the most complex, that its friction resistance was the largest, and that its internal structure was also the most stable of all the curing agents tested. The addition of lime to the curing agents only resulted in the formation of small cracks that reduced overall strength.

## Figures and Tables

**Figure 1 materials-15-06860-f001:**
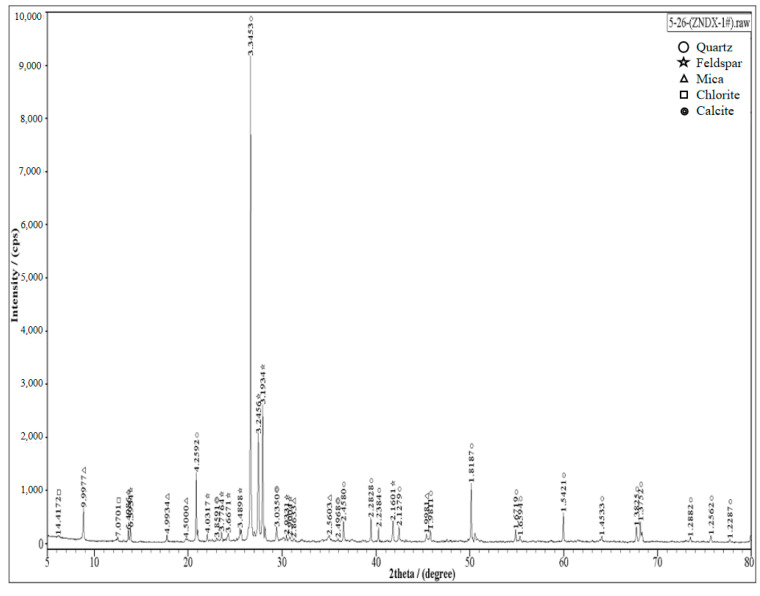
XRD test results.

**Figure 2 materials-15-06860-f002:**
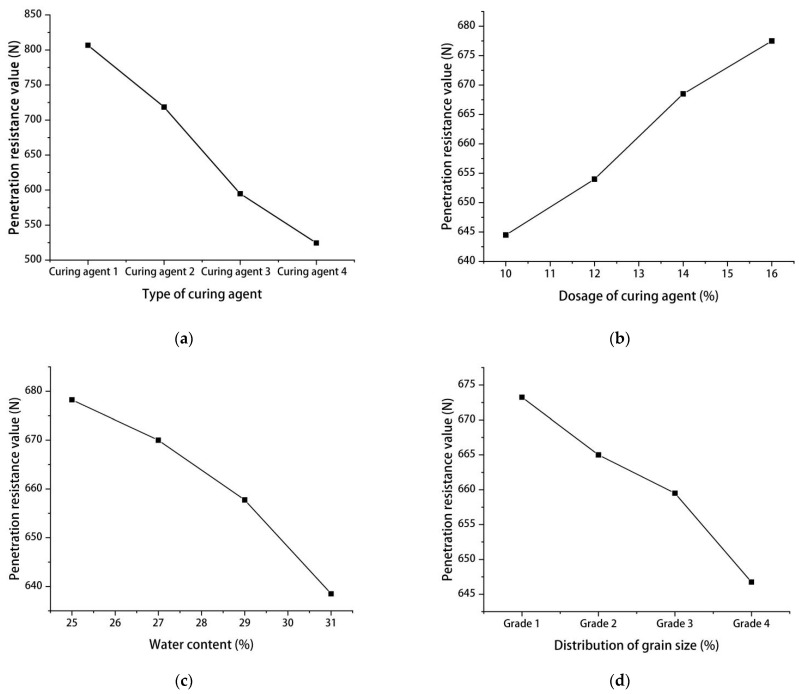
The relationship between each factor and intensity. (**a**): Type of curing agent and strength; (**b**): Curing agent dosage and strength; (**c**): Moisture content and strength; (**d**): Particle size and strength.

**Figure 3 materials-15-06860-f003:**
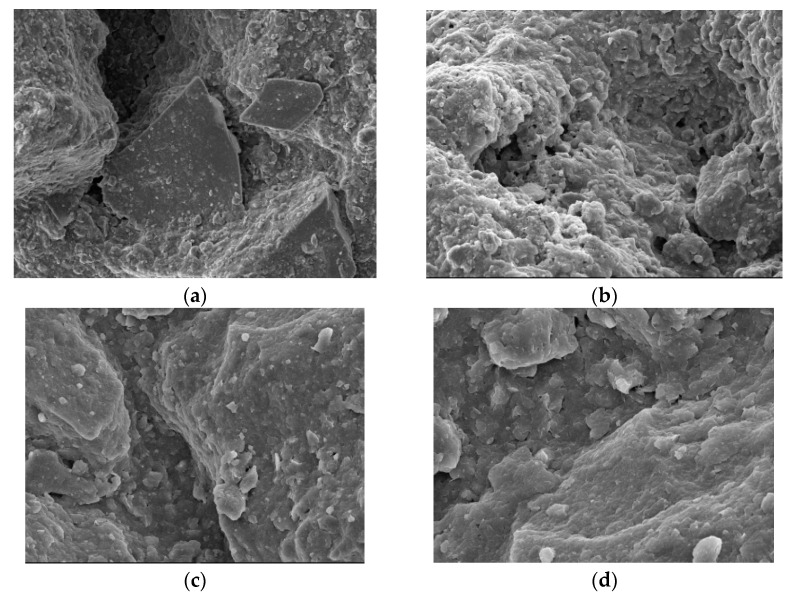
Samples not treated with any curing agent. (**a**): Enlarge the image by 300×; (**b**): Enlarge the image by 500×; (**c**):Enlarge the image by 1000x; (**d**):Enlarge the image by 2000×.

**Figure 4 materials-15-06860-f004:**
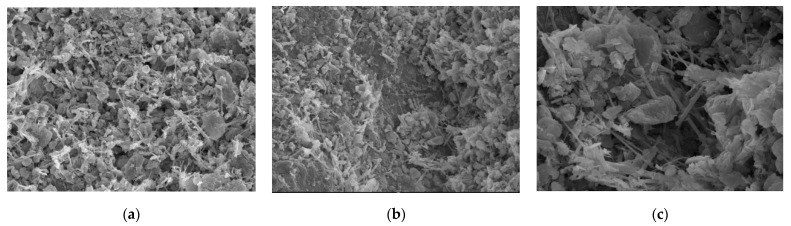
Samples treated with sulphoaluminate cement. (**a**): Enlarge the image by 1000×; (**b**): Enlarge the image by 1000×; (**c**): Enlarge the image by 3000×.

**Figure 5 materials-15-06860-f005:**
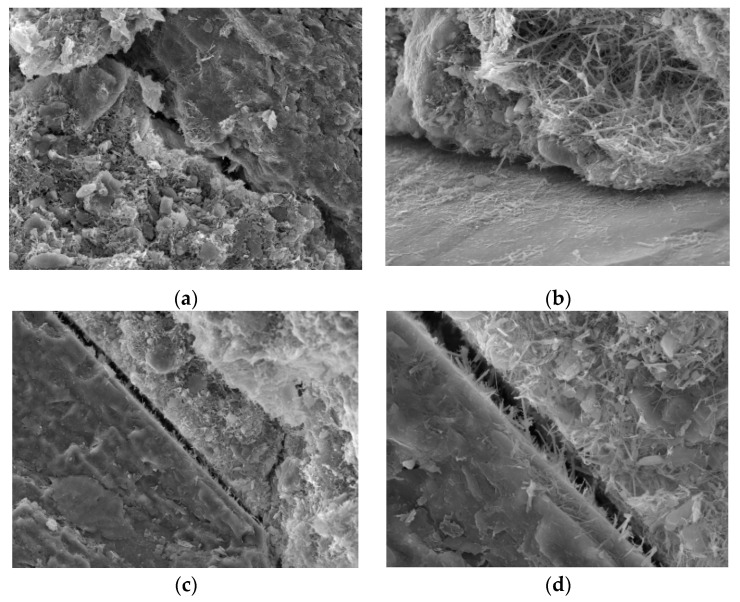
Samples treated with sulphoaluminate cement and lime. (**a**): 1000× magnification; (**b**): 2000× magnification; (**c**): 1000× magnification; (**d**): 3000× magnification.

**Figure 6 materials-15-06860-f006:**
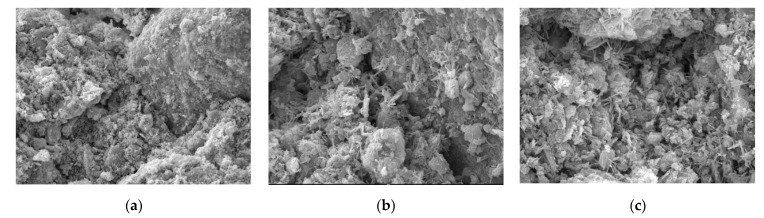
Samples treated with Portland cement. (**a**): 1000× magnification; (**b**): 3000× magnification; (**c**): 3000× magnification.

**Figure 7 materials-15-06860-f007:**
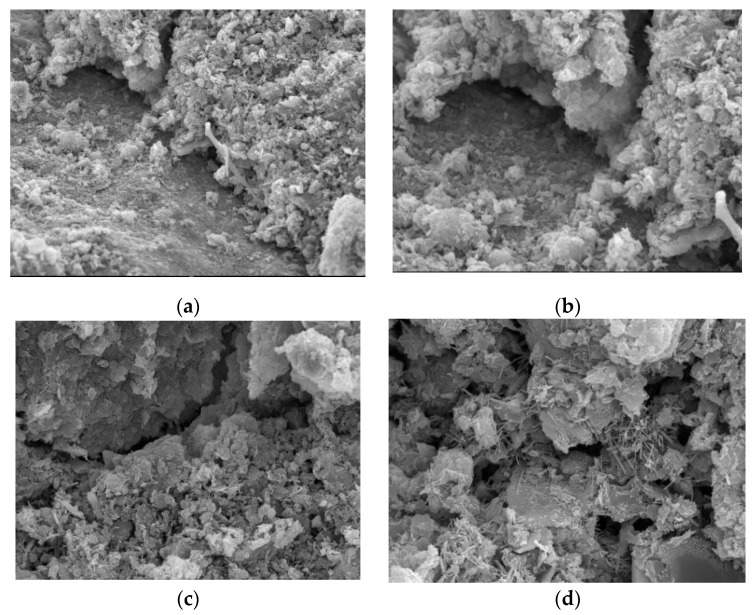
Samples treated with Portland cement and lime. (**a**): 1000× magnification; (**b**): 2000× magnification; (**c**): 2000× magnification; (**d**): 4000× magnification.

**Table 1 materials-15-06860-t001:** Final orthogonal test results.

Serial Numbers	A	B (%)	C (%)	D	Penetration Resistance Value (N)
1	1 (Sulfoaluminate Cement)	1 (10)	1 (25)	1 (Grade 1)	828
2	1	2 (12)	2 (27)	2 (Grade 2)	816
3	1	3 (14)	3 (29)	3 (Grade 3)	805
4	1	4 (16)	4 (31)	4 (Grade 4)	778
5	2 (Sulfoaluminate Cement and Lime (3%))	1	2	3	701
6	2	2	1	4	710
7	2	3	4	1	719
8	2	4	3	2	744
9	3 (Portland Cement)	1	3	4	564
10	3	2	4	3	572
11	3	3	1	2	615
12	3	4	2	1	628
13	4 (Portland cement and lime (0.3%))	1	4	2	485
14	4	2	3	1	518
15	4	3	2	4	535
16	4	4	1	3	560
*k* _1*j*_	3227	2578	2713	2693	-
*k* _2*j*_	2874	2616	2680	2660	-
*k* _3*j*_	2379	2674	2631	2638	-
*k* _4*j*_	2098	2710	2554	2587	-
$\bar{k_{1j}}$	806.75	644.5	678.25	673.25	-
$\bar{k_{2j}}$	718.5	654	670	665	-
$\bar{k_{3j}}$	594.75	668.5	657.75	659.5	-
$\bar{k_{4j}}$	524.5	677.5	638.5	646.75	-
*R*	282.25	33	39.75	26.5	-

**Table 2 materials-15-06860-t002:** Analysis of variance.

Source of Variance	Sum of Squares	Degrees of Freedom	Mean Square	F Value	Significant Results	Critical Value
A	186,669.75	3	62,223.25	485.17	particularly notable	*F*_0.05_ (3, 3) = 9.23
B	1774.25	3	591.42	4.61	have a certain influence	*F*_0.01_ (3, 3) = 29.5
C	5319.10	3	1773.03	13.83	significant	
D	1510.75	3	503.58	3.93	have a certain influence	
Error	384.75	3	128.25			

## Data Availability

The data used to support the findings of this study are included in the article.

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
