# Peer review of "An Experimental Study on the Solidification Treatment of Debris Flow Siltation"

_materials, 2022, doi:10.3390/ma15196860_

Round 1
Reviewer 1 Report
Before the adequate analysis of the results authors have to provide the detailed mineralogical investigation of the studied materials, as different components have different structures and, as the results, the different properties. Please, firstly, provide PXRD results with the quantitative analysis of the components. Moreover, the chemical compositions of the components have to be added.Author Response
Please see the attachment.

Reviewer 2 Report
This is an interesting experimental work on the investigation of solidification of debris flow siltation. The authors conducted mechanical testing, corresponding microstructural characterization, mostly via SEM, and statistical analysis. A list of comments in order to improve this manuscript is below.
1. I suggest that the authors add a short description of the compressive test setup and testing procedures. One of the important parameters to mention would be strain rate.
2. Lines 46-47: I suggest that the authors modify the sentence in this way “Thus, this paper aims to study the solidification treatment of debris flow siltation according to consideration the influence …” or, perhaps, even better “Thus, this paper aims to study the solidification treatment of debris flow siltation by considering the influence …”.
3. Line 57: “dried, crushed and then sieved”.
4. Lines 64-65: “We considered three kinds of curing agents in this study: sulfoaluminate cement …”.
5. The authors indicate that in Table 1 Column E is a blank column. However, column E contains some numbers. I suggest that the authors correct this or clarify their point.
6. Line 89: “We prepared multiple groups of debris flow siltation samples were prepared according to …”. Please, correct the typo.
7. Figure 1: The quality of this figure has to be improved. It is hard to recognize any numbers or any text on this figure.
8. Figures 2-6: At the bottom of each SEM image, there is a narrow horizontal black area with some scanning parameters (I imagine like scale bar, magnification, voltage etc.). The resolution of this part is extremely low and cannot be read. I suggest either removing it from each image or making it clearer.
9. Line 222: Generally, “however” is not used at the end of the sentence.
Reviewer 3 Report
This is a good and important contribution. I have only minor comments summarized below.
Line 28 - "debris flow gullies" - is there a good reference(s) to support this statement?
Line 39 and elsewhere in the text - no need to capitalize the entire surname of first author in the references/
Lines 56-57 - could the authors add some information of the general geologic (field) setting of debris samples used in this study. This will be very helpful in putting this study into a geological context/
Figures 2-6 - please mark various mineral phases directly on the microphotographs. Compositionally different minerals have different physical properties (such as mineral strength, breakability facilitated by cleavage, etc.). These physical properties ay be critical for the initiation of particular mass wasting movement.
Lines 262-265 - what about mineralogy of the debris/waste rock and compositions of individual minerals in the debris flow (see the comment directly above) - do they have any infuence on the curing strength as well as other parameters of a mass wasting event?
Round 2
Reviewer 1 Report
In my opinion, authors should include PXRD patterns as well as discussion of the mineralogical composition into the text.
